# Relationship between Fundamental Movement Skills and Physical Activity in Preschool-aged Children: A Systematic Review

**DOI:** 10.3390/ijerph17103566

**Published:** 2020-05-19

**Authors:** Fei Xin, Si-Tong Chen, Cain Clark, Jin-Tao Hong, Yang Liu, Yu-Jun Cai

**Affiliations:** 1School of Physical Education and Sport Training, Shanghai University of Sport, Shanghai 200438, China; xinhechen1314@gmail.com (F.X.); sussapphire@126.com (J.-T.H.); docliuyang@hotmail.com (Y.L.); 2Institute for Health and Sport, Victoria University, Melbourne 3000, Australia; sitongchen@szu.edu.cn; 3Faculty of Health and Life Sciences, Coventry University, Coventry CV1 5FB, UK; ad0183@coventry.ac.uk; 4Shanghai Research Centre for Physical Fitness and Health of Children and Adolescents, Shanghai University of Sport, Shanghai 200438, China

**Keywords:** fundamental movement skills, physical activity, early years, association

## Abstract

Preschool-aged children are in a critical period of developing fundamental movement skills (FMS). FMS have a close link with physical activity (PA). This study aimed to systematically review the associations between FMS and PA in preschool-aged children. Searching Cochrane Library, Web of Science, PubMed, ScienceDirect, and EBSCO (including SPORTDiscus, ERIC, and Academic Search Premier) was utilized to conduct a systematic review of the available literature. Studies were included if they examined associations between FMS and PA among typically developing children aged 3–6 years, published between January 2000 and April 2020. A total of 26 studies met the inclusion criteria, including 24 cross-sectional studies and two cohort studies. There was a strong level of evidence to support low to moderate associations between moderate to vigorous physical activity and components of FMS, specifically, the total FMS (r = 0.11–0.48, R^2^ = 16%–19%) and object control skill (r = 0.16–0.46, β = 0.28–0.49, R^2^ = 10.4%–16.9%). Similar associations were also found between the total physical activity and components of FMS, specifically, the total FMS (r = 0.10–0.45, R^2^ = 16%), locomotor skills (r = 0.14–0.46, R^2^ = 21.3%), and objective control skills (r = 0.16–0.44, β = 0.47, R^2^ = 19.2%). There was strong evidence that there is no significant association between light physical activity and FMS, specifically, total FMS and locomotor skills. The associations, including “stability skills–PA” and “locomotor skills–moderate to vigorous PA”, were uncertain due to insufficient evidence. Our findings provide strong evidence of associations between specific FMS components and a specific PA intensity. Future studies should consider using a longitudinal study design in order to explore the causal relationship between specific-intensity PA and the FMS subdomain.

## 1. Introduction

Engaging in sufficient physical activity (PA) is linked with a healthy body weight [1,2], cardiorespiratory and muscular fitness [2,3], various cognitive effects [4], and the development of fundamental movement skills (FMS) [5,6] in preschool-aged children. Previous studies have shown that individuals who achieve moderate intensity PA gain more substantial health benefits [7], especially preschool-aged children [8]. The World Health Organization (WHO) recommend that children aged three to six years old participate in at least 60 min of moderate–vigorous physical activity (MVPA) every day [8,9]. However, current evidence suggests that less than half of preschoolers in the USA [10] and Norway [11] meet this recommendation. Consequently, it is imperative to develop strategies to promote PA during early childhood.

Over the past decades, FMS has received a significant amount of attention. FMS are defined as basic learnt movement patterns that do not occur naturally and are suggested to be a foundation for more complex and advanced physical and sporting activities [12]. They can be classified into three categories: locomotor skills (LMS, e.g., running), object control skills (OCS, e.g., catching a ball), and stability skills (SS, e.g., balancing) [12]. It is worth noting that a large number of FMS assessment instruments, such as the Test of Gross Motor Development (TGMD), the CHAMPS (Children’s Activity and Movement in Preschool Study) Motor Skills Protocol (CMSP), evaluate LMS and OCS, but the assessment for SS is usually overlooked. Presently, researchers generally believe that proficiency in FMS is positively associated with being more active in childhood and adolescence [13,14,15]. However, the relationships among type of FMS and PA in early childhood are unclear or inconsistent [16,17], especially regarding the degree and direction of association.

Stodden and his colleagues proposed a conceptual model in 2008 [16], which explains that young children’s PA may drive their development of motor competence. Practice and experience in PA provide various opportunities to stimulate neuromotor development, which, in turn, promotes the development of FMS [16,18]. Some systematic reviews [19,20] have reported that structural PA interventions significantly improve FMS proficiency in preschoolers. Nevertheless, the perspective of Stodden et al. [16] has not been completely documented in the literature. More evidence demonstrates that FMS, as a predictor, can promote PA participation [5,21,22]. The mastery of FMS can be regarded as building blocks of the more specific skills and complex activities [23]. Other cross-sectional studies found that there might be a reciprocal relationship between FMS and PA [24,25,26,27,28]. Children who participate in more MVPA tended to have higher FMS levels, while those with lower FMS levels were observed to spend less time participating in MVPA [24,28]. Intervention studies showed that lessons targeting FMS development may lead to higher PA levels [29]; meanwhile, structured PA might lead to better mastery of FMS [19]. In addition, the relationships among FMS and PA specific to subdomain skills and PA intensity were demonstrated to be different. For the skill aspect, Crane et al. [30] and Iivonen et al. [22] claimed that MVPA had a significant positive correlation with OCS, rather than LMS. However, other studies found that, when compared with OCS, LMS were more strongly related to MVPA [31,32]. In terms of PA, a positive association between FMS and MVPA was found, but there was no relationship between FMS and light physical activity (LPA) [24,33]. Moreover, the relationship became more complicated when the gender variable was added [27,34].

So far, there have only been two systematic reviews [35,36] that have explored the relationship between FMS and PA in preschool-aged children. In 2015, Logan et al. [35] found a low to moderate relationship (r = 0.16–0.48; R^2^ = 3%–23%, *n* = 4) between FMS and PA in young children. OCS and LMS were more strongly related to PA, regardless of gender. Figueroa and An [36] suggested that the specific pattern and degree of the relationship between FMS and PA tended to differ by gender, FMS subdomain, PA intensity, and day of the week (weekdays vs. weekends). However, there is insufficient evidence to support their claims. In summary, these reviews included relatively few studies (*n* = 4 and *n* = 11, respectively) or only synthesized general associations. Current evidence already suggests that there are differences in various dimensions (such as gender, FMS subdomain, and PA intensity) for the relationship between FMS and PA, but few reviews have systematically summarized and analyzed existing evidence. Importantly, clarifying these relationships is not only beneficial for the feasibility of practical implementation, but it also provides a theoretical foundation for teachers, coaches, and therapists to develop courses, training, and interventions. Therefore, the purpose of this study was to systematically review the evidence on the relationship between FMS and PA among preschool-aged children. Based on previous reviews, this review is more comprehensive, including studies from various databases, and evaluates the methodological quality of included studies. In addition, general relationships between FMS and PA, as well as relationships specific to gender, subdomains of FMS, and PA intensity are analyzed.

## 2. Materials and Methods

The process of completing and reporting this review adhered to the Preferred Reporting Items for Systematic Reviews and Meta-analysis (PRISMA) statement guidelines [37].

### 2.1. Search Strategy

A literature search of all electronically archived literature published was conducted in six electronic databases, namely, Cochrane Library, Web of Science, PubMed, ScienceDirect, EBSCO (including SPORTDiscus, ERIC, and Academic Search Premier). Search terms were based on the combination of four parts: (1) preschool* OR kindergarten* OR “early child*” OR “young child*”; (2) (“motor and skill* OR abilit* OR competence OR performance OR proficiency”) OR (“movement and skill* OR abilit* OR competence OR performance OR proficiency”) OR (locomotor OR “object control” OR “manipulat*” OR stability); (3) “physical activit*”; (4) associate* OR relat* OR predict* OR indict* OR corelat* OR effect*. Full-text peer-reviewed articles in English that were published from January 2000 to April 2020 were collected. Appendix A shows the search strategies used in the five databases.

### 2.2. Screen and Selection

After the removal of duplicates, the search results were reviewed and screened by title and abstract with reference to pre-determined inclusion and exclusion criteria, followed by the retrieval of the full texts for evaluation by two authors (FX and JTH). A third author (YL) was consulted to resolve inconsistencies. Studies were selected according to the eligibility criteria.

The inclusion criteria were as follows: the study population had to (1) include typically developing preschool children aged three to six years; (2) objectively assess and report at least one component of fundamental movement skill using a product-oriented or/process-oriented instrument, or both; (3) quantify at least one certain intensity of PA or total PA by a subjective or objective instrument; (4) report associations between FMS and PA and carry out a statistical analysis to report associations including correlations or regression; and, (5) be observational in design.

The exclusion criteria were as follows: (1) the study population was identified as having a pathological condition or disability that affects movement, such as developmental coordination disorder (DCD) or autism spectrum disorder (ASD); (2) participants were infants or were beyond the targeted age range; (3) the study was experimental or review in design; (4) the assessment of motor skills only included fine motor skills or physical fitness; and, (5) the study only reported PA outcomes in terms of PA types (i.e., outdoor PA, recreational PA).

### 2.3. Assessment of Methodological Quality

The methodological quality of cross-sectional studies was assessed by the Agency for Healthcare Research and Quality (AHRQ) scale [38]. Each study was evaluated by eleven items (Ⅰ–Ⅺ, see Table 2) with two responses (yes = 1, no and unclear = 0). If quality assessment scores were from 8 to 11, then the study was considered as high methodological quality. Scores from 4 to 7 were considered as moderate quality, and scores under 3 were considered to be low quality. The methodological quality of cohort studies was assessed using the Newcastle–Ottawa Scale (NOS) [39]. The judgment was based on star scores, with a score of 7 or more stars indicating high quality, 4–6 stars moderate quality, and 0–3 stars low quality. Two authors (FX and JTH) independently assessed the quality of all included studies, and the third author (STC) resolved their disagreements if they existed.

### 2.4. Data Extraction and Synthesis

This systematic review did not conduct a meta-analysis due to the considerable heterogeneity across the included studies [40]. We used a normal data extraction form to collect the following information: first author, publication year, country, participant details, assessment and measurements of PA and FMS, and study findings. The data were extracted by two authors (FX, STC), and inconsistent data were resolved by the third author (YJC). The degree of association was interpreted while using Cohen’s [41] convention as low (r/β = 0.10–0.29), moderate (r/β = 0.30–0.49), or high (r/β ≥ 0.50), and the proportion of shared variance (R^2^) was interpreted as small (R^2^ = 1%–8%), medium (R^2^ = 9%–24%), or large (R^2^ ≥ 25%).

The rule of classification regarding the strength of the relationship between FMS and PA by Sallis et al. [42] was used. The overall level of evidence was categorized into three types: (1) no association, mark “0” (0–33% of studies supporting a significant association); (2) indeterminate/inconsistent association, marked “?” (34–59% of studies and less than four studies supporting a significant association); and, (3) negative or positive association, marked “−” or “+” (≥60% of studies and four or more studies supporting a significant association).

## 3. Results

### 3.1. Study Identification and Characteristics

Figure 1 shows the PRISMA flow diagram, including the systematic literature search, screening, eligibility, and inclusion. Overall, the search initially yielded a total of 1377 potential records. After removing duplicates and screening titles and abstracts, 63 full-text articles were retrieved. Among the 63 full-text studies, there were 26 observational studies (24 cross-sectional studies and two cohort studies) that met the inclusion criteria

Table 1 provides detailed characteristics of the studies included in the systematic review. Of the 26 studies that were included in this review, 23 studies [5,17,21,22,25,26,27,30,31,32,33,43,44,45,46,47,48,49,50,51,52,53,54] were published from 2010 to 2020, while three studies [24,28,34] were published before 2009. Six studies were each performed in the USA [5,28,33,45,51,54], five in the UK [21,24,25,47,52], four in Finland [22,48,49,50], three in Australia [17,34,43], two in Canada [27,30] and Norway [31,32], and one in China [46], Greece [26], Switzerland [53], and South Africa [44]. The sample size of participants varied from 34 [51] to 1081 [31]. A total of 12 studies [5,17,21,22,24,26,28,34,43,46,48,49] applied random sampling.

### 3.2. Methodological Quality of Studies

Agreement rate for the assessment of methodological quality was 88.7% for the 282 items (24 studies × 11 items and 2 studies × 9) between two authors. Disagreements were resolved by discussions until the consensus was reached with the third author. Table 2 and Table 3 show the methodological quality assessment per quality item and per study. In cross-sectional studies, a total of 14 studies [5,17,21,22,24,25,28,30,34,44,46,47,50,53] were categorized to be of high quality, and 10 studies [26,27,31,33,45,48,49,51,52,54] were classified to be of moderate quality. There were no included cross-sectional studies that were classified as low quality. In addition, two cohort [32,43] studies were categorized to be of high quality.

### 3.3. Assessment of FMS and PA

Process-oriented assessments with a qualitative approach were utilized in order to measure FMS in 20 studies. Among these 20 articles, 12 studies applied the TGMD–2 to assess FMS [17,21,25,27,30,34,43,44,47,51,52,54], two studies [28,46] used the CMSP, two studies used the PE Metrics [33,45], one study used the TGMD–3 [5], one study used the Zurich Neuromotor Assessment (ZNA) [53], and two studies used a combination of the TGMD–3 and the Preschooler Gross Motor Quality Scale (PGMQ) [31,32]. The indicators of those assessments included LMS (run, jump, slide, leap, hop, gallop), OCS (throw, catch, roll, kick, dribble, strike), and SS (static/dynamic balance). Six studies used product-oriented assessment with a quantitative approach. FMS were assessed by the Alle kouluikäisten lasten havaintomotorisia ja motorisia perustaitoja mittaavan (APM Inventory) in two studies [22,48] and by a combination of the Koerperkoordinationstest fuer Kinder (KTK) and APM Inventory [49], the Bruininks–Oseretsky Test of Motor Proficiency (BOTMP) [26], the Bruininks–Oseretsky Test Second Edition (BOT–2) [50], and the Movement Assessment Battery for Children (MABC) [24] was used in the other four studies. The APM Inventory, BOTMP, and MABC included LMS, OCS, and SS (balance); BOT–2 included coordination, strength, and agility; the KTK and APM Inventory included OCS and SS; and, the ZNA included LMS and SS.

A total of 22 studies [5,17,21,22,24,25,28,30,31,32,33,34,43,44,45,46,47,49,50,52,53,54] evaluated PA by an accelerometer of which a sampling frequency of 1 to 60 s (including 5, 10, and 15s) was employed and the wear duration ranged from three to 10 days. Two studies [26,51] used a pedometer, and two studies [27,48] used a parents-proxy report questionnaire that was verified by reliability and validity. Three intensities of PA, including LPA, moderate PA, and vigorous PA (VPA), were measured and three main report forms, including LPA, MVPA, and total PA, were used for the results analysis. A total of 19 studies measured and reported MVPA [5,17,21,22,24,25,28,30,33,34,43,44,45,47,48,50,52,53,54], 16 studies measured and reported total PA (TPA) [5,21,22,24,25,26,27,31,32,34,44,46,47,49,51,53], and six studies [24,28,31,32,33,48] reported LPA.

### 3.4. Relationship between FMS and PA

#### 3.4.1. General Relationship between FMS and PA

Overall, 21 [17,21,22,24,25,26,27,28,30,31,32,33,34,43,44,45,46,49,51,53,54] out of the 26 studies showed at least one significant relationship between FMS and PA, regardless of gender, types of skills measured, and PA intensity. To be specific, 16 cross-sectional studies [21,24,25,26,27,28,31,33,34,44,45,46,49,51,53,54] found that FMS had a low to moderate positive correlation with PA (r = 0.10–0.461), while two cross-sectional studies [33,34] found a low to moderate negative correlation (r = −0.18 to −0.50) between FMS and PA. In the regressions of seven studies, six studies [22,30,33,34,51,54] found that FMS was a low to moderate degree predictor for PA (β = 0.20 to 0.49, R^2^ = 10.4%–21.3%). On the contrary, two studies [17,30] found that PA was a moderate degree predictor for FMS (β = 0.37, R^2^ = 10.4%). Two cohort studies [32,43] found that PA at baseline was a significant predictor of FMS at follow up (β = 0.07–0.26), but FMS was not a significant contributor to PA. The strength of associations between FMS types and specific-intensity PA by gender can be found in Table 4.

#### 3.4.2. Gender-Specific Aspects

Only three studies [27,34,49] found that there was a gender-specific relationship between PA and FMS, and the correlations in boys were more positive and stronger than those among girls (r = 0.24–0.55) [27,34,49]. Cliff et al. [34] revealed that only OCS specifically accounted for MVPA in boys (r = 0.48, R^2^ = 0.169), while girls’ LMS was negatively related to MVPA (r= −0.50, R^2^ = 0.192). Total FMS were moderately correlated with MVPA in boys (r = 0.48), whereas there was a moderate negative correlation between total FMS and MVPA in girls (r = −0.46) [34]. Laukkanen et al. [49] found that the correlation between total FMS and TPA was higher in boys (r = 0.448) than in girls (r = 0.138). Temple et al. [27] observed moderate correlations between LMS, OCS, and SS with TPA in boys (r = 0.33–0.41), while these associations did not exist in girls.

#### 3.4.3. Intensity-Specific PA and Total FMS

When compared with single or subdomain FMS, total FMS reflected the overall level of preschoolers’ motor skills. A total of five studies [21,24,28,33,48] examined the relationship between total FMS and LPA. Only one [33] of the studies demonstrated a negative association between total FMS and LPA (r = −0.23); however, non-significant correlations were reported in four studies [21,24,25,28]. Overall, summary coding showed a strong level of evidence supporting the lack of a significant relationship between total FMS and LPA (see Table 5).

The relationship between total FMS and MVPA was examined in 16 studies [5,21,22,24,25,28,33,34,43,44,45,47,48,52,53,54]. Nine [21,24,25,28,33,34,44,45,53] of 11 studies identified a low to moderate positive correlation between total FMS and MVPA (r = 0.114–0.48). Only Tsuda et al. [54] ascertained that the total FMS explained 19% of the variance in MVPA. A total of 12 studies [5,21,22,24,25,26,44,46,47,49,50,53] examined the bivariate relationship regarding the TPA and total FMS. Eight [21,24,25,26,44,46,49,53] of nine studies demonstrated a low to moderate correlation between total FMS and TPA (r = 0.10–0.448). In addition, the regression results of Iivonen et al. [22] showed that the total FMS explained 16% of the variance in MVPA and TPA, respectively. Overall, summary coding presented a strong level of evidence supporting low to moderate positive associations among MVPA, TPA, and total FMS (see Table 5).

#### 3.4.4. Intensity-Specific PA and Subdomain Skills

In terms of LPA, only Gu [33] found that the assessment of LMS (r = −0.18, β = −0.19) and OCS by TGMD−2 identified weak negative correlations with LPA (r = −0.18). Several other studies showed non-significant correlations between LMS [21,28,31,32], OCS [28,31,32], and SS [31,32] and LPA. Overall, summary coding presented a sufficient level of evidence in order to support the presence of a non-significant relationship between LMS and LPA, while the level of current evidence to support non-significant relationships among OCS, SS, and LPA was uncertain (see Table 5).

As for MVPA, a total of 17 studies examined the relationship between MVPA and FMS subdomains (including LMS [5,17,22,25,28,30,31,32,33,34,43,44,45,47,52], OCS [5,17,22,25,28,30,31,32,33,34,43,44,45,47,52,54], and SS [22,31,32]). Five studies [28,31,33,44,45] found weak positive correlations among LMS (r = 0.16–0.26), OCS (r = 0.016–0.25), and MVPA. In a study by Gu [33], only LMS could explain variance in MVPA to a low degree (β = 0.20), but Crane et al. [30] found that only OCS was a significant contributor to MVPA (β = 0.281). In the study of Tsuda et al. [54], there was a higher correlation between MVPA and LMS (r = 0.53) when compared with OCS (r = 0.46). Conversely, Hall et al. [25] stated identified a significant relationship between MVPA and OCS (r = 0.367) but not with LMS. Another two studies [22,34] found that both LMS (β = 0.35, R^2^ = 0.192) and OCS (β = 0.49, R^2^ = 0.169) had moderate predictive roles in MVPA. In addition, two studies [17,30] claimed that MVPA only had a moderate predictive role in OCS (β = 0.37, R^2^ = 0.104), while MVPA had no significant predictive role in LMS. In only two cohort studies, MVPA at baseline was found to be a significant predictor of LMS (β = 0.23), OCS (β = 0.22), and SS (β = 0.17) [32]. Finally, when comparing with LMS and OCS, the association between SS and MVPA in two studies was identified as not being significant [22,31]. Overall, summary coding presented that there was a strong level of evidence to support a relationship between OCS and MVPA, but the association between LMS and MVPA was inconsistent, as less than 60% of studies supported this relationship. In addition, the relationship between SS and MVPA was uncertain due to a lack of sufficient studies (see Table 5).

Regarding TPA, 11 studies examined the relationship between the FMS subdomains (including LMS [5,22,25,26,27,31,44,46,47,51], OCS [5,22,25,26,27,31,34,44,46,47,51], and SS [22,26,27,31]) and TPA. Three studies [31,44,46] demonstrated low positive correlations between LMS (r = 0.14–0.24), OCS (r = 0.16–0.20), and TPA. In the study of Temple et al. [27], TPA was found to be significantly associated with LMS to a low degree (r = 0.24), but it was not associated with OCS. Five studies [22,25,26,34,51] claimed that OCS was significantly correlated with TPA (r = 0.30–0.435, β = 0.47, R^2^ = 19.2%), while only two studies [26,51] discovered a moderate association between LMS and TPA (r = 0.31–0.461, R^2^ = 21.3%). Few studies have been conducted on the relationship between SS and PA. Kambas et al. [26] stated that dynamic balance was positively associated with aerobic walking time (r = 0.401), while the other three studies found that SS had no significantly predictive effect on TPA [22]. Overall, summary coding presented a strong level of evidence to support relationships among LMS, OCS, and TPA, but the association between SS and MVPA was uncertain, due to a lack of sufficient studies (see Table 5).

## 4. Discussion

The aim of this study was to systematically review the associations between aspect of FMS and PA in preschool-aged children. This study is the first review to synthesize and analyze the associations among specific types of FMS and specific intensities of PA in young children. A total of 26 observational studies met the eligible criteria, including 24 cross-sectional studies and two cohort studies. Sixteen studies were evaluated to be of high methodological quality and 10 studies were determined to be of medium quality. In summary, there was strong level of evidence to support a low to moderate positive association between FMS (specifically total FMS and OCS) and MVPA in preschool-aged children, while there was an inconsistent association between LMS and MVPA. Sufficient evidence supported a low to moderate association between FMS (specifically, total FMS, LMS, and OCS) and TPA. There was also sufficient evidence to support the lack of significant associations among LMS, total FMS, and LPA, and a similar trend of no significant association between OCS and LPA was demonstrated, although the evidence that was presented in the included studies was relatively insufficient. Finally, this review could not identify a significant or non-significant association between SS and any intensity of PA due to a lack of sufficient evidence. Regarding the gender-specific aspect, the difference in associations between boys and girls requires further evidence to be clarified. Our findings are important, because understanding specific relationships between FMS and PA may be significant for improving specific skills or a specific PA intensity.

Regardless of gender, this review found a strong level of evidence to support low to moderate positive associations between the total TPA and components of FMS, specifically LMS, OCS, and total FMS. This finding is consistent with the conclusion of Logan et al. (r = 0.16 to 0.48; R^2^ = 3%–23%, *n* = 4) [35]. This evidence generally verifies the hypothesis of Stodden et al., who developed a conceptual model showing that the association between variable levels of PA and FMS is weak in the early childhood period [16]. However, we have insufficient evidence to support another hypothesis from Stodden et al., which stated that PA might drive the development of motor skill competence. Although one cohort study ascertained that the total TPA at baseline was a significant predictor of total FMS, we need more longitudinal studies in order to verify this assumption. We tried to speculate the reasons for the low to moderate associations between total FMS and TPA while using the perspective of multidimensional factors. Using the ecological prospective, there are various significant factors related to PA and FMS, such as the immediate environment, structured physical education, parental influence, and climate. Welk [55] categorized determinants of PA into five aspects—personal, biological, psychological, social, and environmental—where motor skills were only a factor at the biological level of PA. Niemistö et al. [56] also identified that individual (age, body mass index, temperament), family (parents’ education level), and environmental (access to sports facilities) factors were associated with motor skills development in children. Another probable explanation is that the key role of mediating variables (e.g., perceived motor competence, health–related physical fitness, and obesity) might decline/promote or interact with the relationship between FMS and PA [16,57,58]. Only two [30,47] of the included studies explored this mediating effect among preschoolers, but the results were not significant. One relevant explanation for this is that young children tend to exaggerate their perceived competence level relative to their actual motor competence, because they are unable to make social comparisons, differentiate between actual and real self-concept, and take the perspective of others [59]. However, although perceived competence in young children was considered to be of limited accuracy, inflated perceived competence could improve motor skills to some extent, because they were sure that they would be skillful. It is necessary to determine the roles of perceived motor competence and other mediating factors in future studies in order to clarify the mechanism of the relationship and broaden the scope of motor skills. In general, positive but weak associations in early childhood certainly support an emerging developmental relationship.

Regarding the gender-specific aspect, the differences in associations between components of FMS and specific intensities of PA in boys and girl were uncertain due to limited evidence. Evidence in cross-sectional studies suggested that boys are more physically active than girls [10,11,60] and have greater OCS [5,21,27,61], whereas girls’ LMS were usually higher than those of boys [5,34,61]. Therefore, it is necessary to add evidence of gender differences from the perspective of specific gender interventions. The included studies comparing this association provided inconsistent conclusions. Some studies indicated moderate associations between LMS, OCS, and TPA [34], OCS and MVPA for boys [34], while other studies suggested a non-significant [27,49] or negative association between FMS and PA among girls [30]. On the one hand, FMS probably plays a more prominent role in the prerequisites for PA participation in boys [62]; in other words, boys are keener to participate in skill-oriented activities. On the other hand, the stronger relationship is partly due to the preferential type of OCS assessment, which, for boys, was ball sports (e.g., soccer, basketball, cricket, baseball) [63] in most of the current studies. However, for girls, the results showed no association of FMS with PA due to the lack of balance, rhythm, and preferred skills in assessments. Negative associations in girls were unusual, and this might have been caused by sample limitations (e.g., sample size, age, demographics) in the investigations [34]. Most of the current studies did not compare gender differences in the relationship and this observation might be due to the use of a single type of global movement skills score rather than using categories for the movement skills scores (for LMS, OCS, or SS) [24]. The lack of gender difference might be related to the age and the stage of cohort, as preschool-aged children are in early childhood and their FMS have not yet matured. Relevant evidence needs to be supplemented by more research in the future, given the importance of gender-targeted interventions.

There was a strong level of evidence to support a low to moderate positive association between FMS (specifically total FMS and OCS) and MVPA in preschool-aged children, while there was a inconsistent association between LMS and MVPA. The WHO proposed PA guidelines for children under five years old for the first time, which recommend that children should spend at least 60 min participating in MVPA per day [8]. From the perspective of PA promotion, it would be worthwhile to explore the respective contributions of OCS and LMS to MVPA. The study of Hall et al. [25] suggested that children who achieved ≥60 min of MVPA had significantly better OCS scores. Similar to previous studies in middle childhood [13,64], it appears that OCS, rather than LMS, is associated with high levels of MVPA in early childhood [17,25,30]. There is sufficient evidence regarding why MVPA is more closely related to OCS, but not LMS. It has been suggested that the ball skills component of OCS is fundamental to participation in various games and sports that involve OCS (i.e., soccer or baseball) [13]. Barnett et al. [13,57] found that the performance of OCS in childhood, as well as in adolescence, was significantly associated with time spent in MVPA in adolescence. The similar explanation in this review demonstrates that the importance of OCS manifests in preschool-aged children. The evidence provided in this review suggested suggests that there are non-significant or even negative correlations between LPA and components of FMS, specifically total FMS and LMS. The evidence supporting the lack of an association between OCS and LPA is insufficient. It is possible that this uncertain association was caused by researchers not reporting result data or not including the variable of LPA. In terms of TPA, it seems that LMS and OCS are equally positively related to TPA.

The use of different types of PA (subjective vs. objective) and FMS assessment tools (product-oriented vs. process-oriented) might have a crucial impact on the comparability of study results. Therefore, these factors should be interpreted and discussed carefully. Accelerometers are as the “gold standard” for objective measurements, and they have been proven to be reliable and valid when used in young children [65]. The most included studies measured PA by accelerometers. However, the different sampling frequencies and cut points used in studies [24,30] affected the classification of PA intensity, which may have led to the underestimate of multidirectional and sporadic PA and was not conducive to comparisons among study results. Two studies [26,51] used pedometers, which is, they measured PA by the number of steps and they could not classify PA. Nevertheless, when compared with objective measurements, subjective instruments (e.g., parent-proxy questionnaires [27,48], and diaries) can be used to measure different PA types, such as recreational activities and skill-based activities, but their results tended to be overestimated or even inaccurate. Crane et al. [30] suggested that asking parents to complete a detailed physical activity log by observation to accompany the accelerometer data would go some way to unpacking this relationship. In fact, it is largely insufficient if studies just measure the amount of different intensities of PA completed in discrete time periods. Some intervention studies found that structural or organized PA could significantly improve FMS competence. Hence, studies examining this association should make a comparison between organized and non–organized PA and they should even access skill-related PA.

The relationship between FMS competence and PA behaviors might be influenced by the orientation of the assessment tool. Webster et al. [5] determined that total FMS and LMS scores assessed by TGMD–3 (process-oriented) were associated with more vigorous physical activity time. In contrast, there was no significant association between PA and FMS measured by MABC (product-oriented). The performance of process- and product-oriented assessments of FMS differs across skills and age groups (r = 0.26–0.88) [66], which complicates the comparison of results. In process-oriented assessments, the comparison of a child with an “expert” performer often results in ceiling effects and floor effects, which reduces the validity. In product-oriented assessments, product measurements do not observe the developmental movement process related to the movement product [16]. Notably, process-oriented assessments seem to get more attention and have greater use by researchers. The series of TGMD tools developed by Ulrich [67,68] is one of the most widely used assessments of FMS, but it does not contain SS, which is regarded as an indispensable component of FMS. The CMSP and PE Metrics also do not involve SS. Of course, there are some assessments measuring not only LMS and OCS, but also SS, such as Motorische Basiskompetenzen (MOBAK) [69], PGMQ [70]. In addition to measuring locomotor and object control skills with TGMD–2, Nilsen et al. [32] also measured dynamic and static balance using the PGMQ. Breaking the limitations of single assessments and combining scores from different assessments might improve scientific evaluation. These observations highlight the necessity of adopting a multidimensional method to understand FMS in preschoolers and combining both product- and process-oriented assessments to deepen the understanding of the delicate differences in the relationship between FMS and PA.

Our systematic review has several critical strengths. Firstly, this review performed a more in-depth analysis of associations between different types of FMS and specific intensities of PA in preschool-aged children than previous reviews. These specific associations are critical for providing a reference for the experiment design of interventions. Secondly, the methodological quality of all included studies was medium to high, which gave rise to reliable evidence. Thirdly, the strength of the evidence was classified in this review to provide more dependable evidence. This study also has some limitations. Firstly, we did not take mediators of the association between FMS and PA into account, although a few studies did analyze the effect of mediators in the association. Secondly, a meta-analysis was not conducted due to the lack of a uniform effect size and adequate studies. In addition, few studies took the influences of other variables, such as demographics, social-economic status, body mass index, and race into account. Therefore, we solely examined the relationship between FMS and PA.

## 5. Conclusions and Implication

This systematic review provides strong evidence of the rapidly increasing amount of literature examining associations between FMS and PA. The review found a strong level of evidence to support a low to moderate positive association between FMS (specifically total FMS and OCS) and MVPA in preschool-aged children, whereas there was an inconsistent association between LMS and MVPA. There was sufficient evidence that TPA is significantly and positively associated with components of FMS (specifically total FMS, LMS, and OCS) to a low to moderate degree. There was also sufficient evidence to support the lack of significant associations among LMS, total FMS, and LPA; meanwhile, a similar trend of no significant association was demonstrated between OCS and LPA, although the evidence of included studies was relatively insufficient. However, due to lack of sufficient evidence, this review could not identify a significant or non-significant association between SS and any intensity of PA. In addition, the differences in associations between boys and girl were uncertain due to lack of sufficient evidence. The general findings of this review should be interpreted with caution due to the use of different populations from various cultural backgrounds, the diversity and complexity of the assessment, and the varying levels of methodological quality.

Evidence from cohort studies examining cause and effect relationships was insufficient, but these findings were valuable for examining the intervention study design, especially regarding the assessment of different components of FMS and the specific intensity of PA based on gender. In future cross-sectional studies, it is necessary to add studies examining the association between LMS and MVPA and the association between SS and the specific intensity of PA. Researchers need to utilize a more uniform or mixed assessment of FMS that include not only LMS and OCS but also SS in order to provide more accurate and comprehensive findings in the future. Evidence suggests that accelerometers may underestimate the intensity of PA in manipulative skill performance, and we recommend the use of more scientific sampling frequencies and accelerometer cut off points in future studies.

## Figures and Tables

**Figure 1 ijerph-17-03566-f001:**
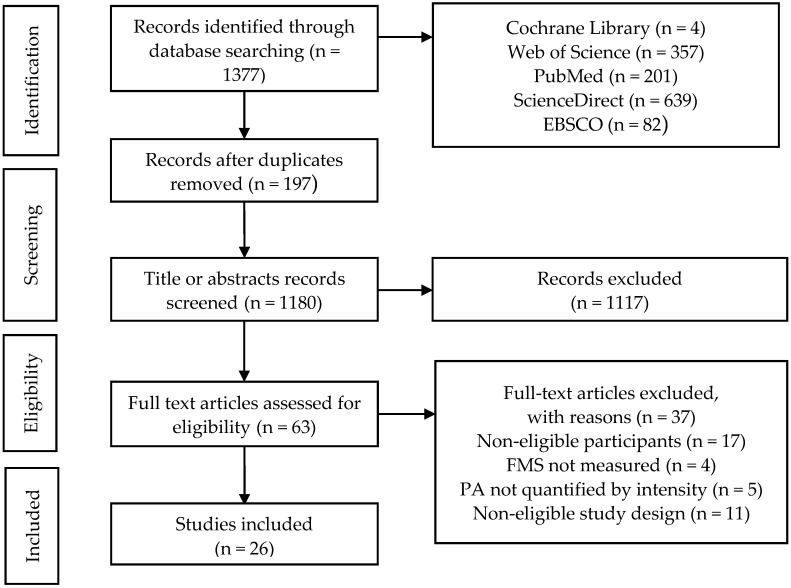
The Process of Articles Retrieve.

**Table 1 ijerph-17-03566-t001:** General characteristics of the studies included in the systematic review.

First Author Year Country	Participant Details	Assessment		Study Findings
		PA	FMS	
Barnett 2013 Australia [17] ^C^	76 (42 girls) 4.1 ± 0.68 y	Accelerometer (15 s) At least 4 days (3 weekdays and 1 weekend day)	TGMD−2 ^+-^	Regressions: moderate–vigorous physical activity (MVPA) was a sig. predictor of object control skills (OCS), β = 0.37 *, but was not a sig. predictor of locomotor skills (LMS).
Barnett 2016 Australia [43] ^C,L^	127 (68 girls) 5.0 ± 0.1 y 118 (65 girls) 3.5 ± 0.2 y	Accelerometer (15 s) At least 4 days (3 weekdays and 1 weekend)	TGMD−2 ^+^	Regressions: MVPA at 3.5 years was a predictor of LMS at age 5 years, β = 0.073*. MVPA at 5 years was not associated with LMS. MVPA was not associated with OCS fundamental movement skills (FMS) at any age.
Cliff 2009 Australia [34] ^C^	46 (21 girls) 4.3 ± 0.7 y	Accelerometer (60 s) At least 3 days	TGMD−2 ^+^	Correlations for boys: MVPA and total FMS r = 0.48 *, & OCS r = 0.48 *, & LMS, not sig.; for girls: MVPA & GMQ r = −0.46 *, & LMS r = −0.50 *, & OCS not sig. Regressions for boys: OCS explained 16.9% and 13.7% of var. in MVPA and total physical activity (TPA), respectively; for girls: LMS explained 19.2% of var. in MVPA.
Cook 2019 South Africa [44] ^C^	129 (68 girls) 4.2 y	Accelerometer (15 s) At least 3 days	TGMD−2 ^+^	Correlations: LMS & MVPA r = 0.18, & TPA r = 0.14; OCS & MVPA r = 0.25, & TPA r = 0.20; FMS & MVPA r = 0.24, & TPA r = 0.15.
Crane 2015 Canada [30] ^C^	116 (49 girls) 5.6 y	Accelerometer (15 s) At least 4 days (3 weekdays and 1 weekend day)	TGMD−2 ^+^	Regressions: OCS were sig. predictors of MVPA, β = 0.281 **, LMS were not a sig. predictor of MVPA. MVPA explained 10.4% of var. in OCS, but MVPA was not a sig. predictor of LMS.
Fisher 2005 UK [24] ^C^	394 (185 girls) 4.2 ± 0.5 y	Accelerometer (60 s) 6 days	MABC ^-^	Correlations: total FMS & TPA r = 0.10 * & MVPA r = 0.18 **; & LPA not sig.
Foweather 2015 UK [21] ^C^	99 (47 girls) 4.6 ± 0.5 y	Accelerometer (5 s) 7 days	TGMD−2 ^+^	Correlations: FMS & MVPA r = 0.114 **, & TPA r = 0.102 **.
Gu 2016 USA [33] ^C^	256 (127 girls) 5.37 ± 0.48 y	Accelerometer (60 s) 5 school days	PE Metrics ^+^	Correlations: LMS & LPA r = −0.18 **, MVPA r = 0.21 **; OCS & LPA r = −0.18 **, MVPA r = 0.21 **; total FMS & LPA r = −0.23 **, MVPA r = 0.26 **. Regressions: LMS was a sig. predictor of LPA, β = −0.19, and MVPA, β = 0.20; OCS was not a sig. predictor of MVPA.
Gu 2017 USA [45] ^C^	141 (69 girls) 5.37 ± 0.48 y	Accelerometer (60 s) 5 school days	PE Metrics ^+^	Correlations: LMS & MVPA r = 0.24 **; OCS & MVPA r = 0.24 **; FMS & MVPA r = 0.28**.
Guo 2018 China [46] ^C^	227 (116 girls) 4.15 ± 0.63 y	Accelerometer (15 s) At least 3 days	CMSP ^+^	Correlations: TPA & LMS r = 0.14 *, & OCS r = 0.16 *; & total FMS r = 0.17 *. Regressions: LMS was a sig. predictor of TPA β = 0.04 *, while OCS and total FMS were not sig. predictors of TPA.
Hall 2018 UK [25] ^C^	166 (75 girls) 4.28 ± 0.74 y (3–5 y)	Accelerometer (1 s) 4 days (2 weekdays and 2 weekend days)	TGMD−2 ^+^	Correlations: total FMS & MVPA r = 0.376 **, & TPA r = 0.402 **; OCS & MVPA r = 0.367 **, & TPA r = 0.386 **; LMS & MVPA, & TPA not sig.
Hall 2019 UK [47] ^C^	38 (14 girls) 5.37 ± 0.79 y (4–6 y)	Accelerometer (60 s) 4 days (2 weekdays & 2 weekend days)	TGMD−2 ^+^	Regressions: FMS (total FMS, LMS, OCS) were not sig. predictors of physical activity (PA) (MVPA, TPA) and vice versa.
Iivonen 2013 Finland [22] ^C^	37 (20 girls) 4.1 ± 0.34 y	Accelerometer (5 s) 5 days (at least 2 weekdays and 1 weekend day)	APM ^-^	Regressions: total FMS was a predictor of MVPA, β = 0.49 **, and TPA, β = 0.49 **; OCS was a sig. predictor of MVPA, β = 0.49 **, and TPA, β = 0.47 **; LMS was a sig. predictor of MVPA, β = 0.35 **, but not a sig. predictor of TPA; stability skills were not sig. predictors of MVPA and TPA.
Iivonen 2016 Finland [48] ^C^	53 (29 girls) 4.07 ± 0.32 y	OSRAC–P	APM ^-^	Regressions: LPA and MVPA were not sig. predictors of total FMS.
Kambas 2012 Greece [26] ^C^	232 (114 girls) 5.37 ± 0.28 y	Pedometer 7 days	BOTMP ^-^	Correlations: TPA & LMS r = 0.31–0.39 *, & OCS r = 0.30–0.35 *, & SS r = 0.401 *, & FMS r = 0.368
Laukkanen 2014 Finland [49] ^C^	53 (28 girls) 5.94 y	Accelerometer (15 s) 6 days (at least 2 weekdays and 1 weekend day)	KTK & APM manipulative skill test ^-^	Correlations for boys: total FMS & TPA r = 0.448 *;for Girls: total FMS & TPA r = 0.138.
Matarma 2018 Finland [50] ^C^	111 (66 girls) 5.57 ± 0.4 y	Accelerometer (15 s) 7 days (at least 3 weekdays and 1 weekend day)	BOT−2 ^-^	Regressions: total FMS was not a sig. predictor of MVPA.
Nilsen 2020 Norway [31] ^C^	1081 (526 girls) 4.7 ± 0.9 y	Accelerometer (1 s) At least 4 days (3 weekdays and 1 weekend day)	TGMD−3 ^+^ PGMQ ^+^	Correlations: LPA & LMS, & OCS, & SS, not sig; MVPA & LMS, r = 0.26, & OCS r = 0.16, & SS not sig.; TPA & LMS, r = 0.23, & OCS r = 0.16, & SS not sig.
Nilsen 2020 Norway [32] ^L^	230 (110 girls) 4.7 ± 0.9 y	Accelerometer (1 s) At least 4 days	TGMD−3 ^+^ PGMQ (SS) ^+^	Regressions: MVPA at baseline was sig. predictor of LMS, β = 0.26 **, OCS, β = 0.18, and SS, β = 0.19 at follow up; TPA at baseline was a sig. predictor of LMS, β = 0.23 **, OCS, β = 0.22, and SS, β = 0.17 at follow up; LPA was not a sig. predictor of FMS. FMS at baseline was not a sig. predictor of PA at follow up.
Robinson 2012 USA [51] ^C^	34 (22 girls) 4.75 ± 0.53 y	Pedometer 3 school days	TGMD−2 ^+^	Correlations: LMS & TPA r = 0.461 *, OCS & TPA r = 0.435 *. Regressions: LMS explained 21.3% of the var. in TPA;OCS was not a sig. predictor of TPA.
Roscoe 2019 UK [52] ^C^	185 (86 girls) 3.4 ± 0.5 y	Accelerometer (10 s) 4 days (2 weekdays and 2 weekend days)	Adapted TGMD−2 ^+^	Correlations: No sig. corr. between FMS (total FMS, LMS, OCS) and MVPA.
Schmutz 2020 Switzerland [53] ^C^	550 (292 girls) 3.9 ± 0.7 y	Accelerometer (15 s) At least 3 days (including 1 weekend day)	ZNA	Correlations: FMS & MVPA r = 0.23, & TPA r = 0.24
Temple 2016 Canada [27] ^C^	74 (33 girls) 5.92 ± 0.33 y	CAPE	Stork stand ^-^ TGMD−2 ^+^	Correlations: TPA & LMS r = 0.24 *, & OCS, & SS not sig.; for boys: TPA & LMS r = 0.41 **, & OCS r = 0.40 *, & SS r = 0.33 *; for girls: TPA & LMS, & OCS, & SS not sig.
Tsuda 2019 USA [54] ^C^	72 (33 girls) 4.38 ± 0.85 y	Accelerometer (15 s) 3 school days	TGMD−2 ^+^	Correlations: LMS & MVPA r = 0.53 **; OCS & MVPA r = 0.46 **. Regressions: LMS and OCS explained 19% of var. in MVPA.
Webster 2018 USA [5] ^C^	126(68 girls) 3.4 ± 0.5 y	Accelerometer (15 s) 7 days (at least 3 days)	TGMD−3 ^+^	Regressions: FMS (total FMS, LMS, OCS) were not sig. predictors of PA (MVPA, TPA).
Williams 2008 USA [28] ^C^	198 (98 girls) 4.2 ± 0.5 y	Accelerometer (15 s) 8–10 days	CMSP ^+^	Correlations: FMS & LPA not sig, & MVPA r = 0.20 **; LMS & LPA not sig, & MVPA r = 0.16 *; OCS & LPA not sig, & MVPA r = 0.19 *.

Notes: ^C^: cross-sectional study; ^L^: cohort study; y: years; ^+^: process–oriented assessment; ^-^: product–oriented assessment; TGMD: Test of Gross Motor Development; MABC: Movement Assessment Battery for Children; CMSP: CHAMPS Motor Skill Protocol; APM Inventory: Alle kouluikäisten lasten havaintomotorisia ja motorisia perustaitoja mittaavan; BOTMP: Bruininks-Oseretsky Test of Motor Proficiency; PGMQ: Preschooler Gross Motor Quality Scale; KTK: Koerperkoordinationstest fuer Kinder; LMS: locomotor skill; OCS: object control skill; FMS: fundamental movement skill; OSRAC–P: Observational System for Recording PA in Children–Preschool Version; CAPE: Children’s Assessment of Participation and Enjoyment survey; LPA: light physical activity; MVPA: moderate-vigorous physical activity; TPA: total physical activity; β: standardized regression coefficient; corr.: correlation; *: *p* < 0.05; **: *p* < 0.01; r: correlation coefficient; R^2^: coefficient of determination; sig: a significant association. var.: variance.

**Table 2 ijerph-17-03566-t002:** Methodological quality assessment of included cross-sectional studies.

Study	Criteria											Total Score
	I	II	III	IV	V	VI	VII	VIII	IX	X	XI	
Barnett 2013	1	1	1	1	1	1	1	1	0	1	0	9
Cliff 2009	1	1	1	1	1	0	1	1	1	1	0	9
Cook 2019	1	1	1	1	1	0	1	1	1	1	0	9
Crane 2015	1	1	1	1	1	0	1	1	0	1	0	8
Fisher 2005	1	1	1	1	1	1	1	0	1	1	0	9
Foweather 2015	1	1	1	1	1	0	1	1	1	1	0	9
Gu 2016	1	1	1	1	1	1	1	1	1	1	0	10
Gu 2018	1	0	1	1	1	1	0	1	1	0	0	7
Guo 2018	1	1	1	1	1	1	0	1	0	0	0	7
Hall 2018	1	1	1	1	1	1	1	1	0	1	0	9
Hall 2019	1	1	1	1	1	1	1	1	0	1	0	9
Iivonen 2013	1	1	1	1	1	1	1	1	0	0	0	8
Iivonen 2016	1	0	1	1	1	0	0	0	0	1	0	5
Kambas 2012	1	1	1	1	1	0	1	0	0	1	0	7
Laukkanen 2014	1	1	1	1	1	0	1	1	0	0	0	7
Matarma 2018	1	1	1	1	1	0	1	1	0	1	0	8
Nilsen 2020 b	1	0	1	1	1	1	0	1	0	1	0	7
Robinson 2012	1	0	1	1	1	1	0	0	0	0	0	5
Roscoe 2019	1	0	1	1	1	1	1	0	0	0	0	6
Schmutz 2020	1	1	1	1	1	1	1	1	1	1	1	11
Temple 2016	1	0	1	1	1	0	0	1	0	1	0	6
Tsuda 2019	1	0	1	1	1	1	0	1	0	0	0	6
Webster 2018	1	1	1	1	1	1	0	1	0	1	0	8
Williams 2008	1	1	1	1	1	1	1	1	1	1	0	10

Notes: Ⅰ: Define the source of information (survey, record review); Ⅱ: List inclusion and exclusion criteria for exposed and unexposed subjects (cases and controls) or refer to previous publications; Ⅲ: Indicate time period used for identifying patients; IV: Indicate whether or not subjects were consecutive if not population-based; V: Indicate if evaluators of subjective components of study were masked to other aspects of the status of the participants; VI: Describe any assessments undertaken for quality assurance purposes (e.g., test/retest of primary outcome measurements); VII: Explain any patient exclusions from analysis; VIII: Describe how confounding was assessed and/or controlled; IX: If applicable, explain how missing data were handled in the analysis; X: Summarize patient response rates and completeness of data collection; XI: Clarify what follow-up, if any, was expected and the percentage of patients for which incomplete data or follow-up was obtained.

**Table 3 ijerph-17-03566-t003:** Methodological quality assessment of included cohort studies.

Study	Criteria									Total Score
	1a	1b	1c	1d	2a	2b	3a	3b	3c	
Barnett 2016	*	*	*	*	*	*	*	*		8
Nilsen 2016	*	*	*	*	*	*	*	*	*	9

Notes: 1a: Representativeness of the exposed cohort; 1b: Selection of the non-exposed cohort; 1c: Ascertainment of exposure; 1d: Demonstration that outcome of interest was not present at start of study; 2a: Cohorts comparable on basis of age; 2b: Cohorts comparable on other factor(s); 3a: Assessment of outcome; 3b: Follow-up was long enough for outcomes to occur; 3c: Adequacy of follow up of cohorts. *: one star could be awarded for the item if the study met the specific quality criterion.

**Table 4 ijerph-17-03566-t004:** The strength of associations between fundamental movement skills (FMS) types and specific-intensity physical activity (PA) by gender.

	Not Sig			Low Level			Moderate Level			Strong Level		
	Boys	Girls	Total	Boys	Girls	Total	Boys	Girls	Total	Boys	Girls	Total
FMS-LPA			[21,24,28,48]			[33] −						
LMS-LPA			[21,28,31,32]			[33] −						
OCS-LPA			[28,31,32]			[33] −						
SS-LPA			[31,32]									
FMS-MVPA			[5,43,47,48,52]			[21,24,28,33,44,45,53]	[34]	[34]-	[22,25,54]			
LMS-MVPA	[34]	[34]	[5,17,25,30,43,47,52]			[28,31,32,33,44,45]			[22,34]			[54]
OCS-MVPA		[34]	[5,33,43,47,52]			[28,30,31,32,33,44,45]	[34]		[17,22,25,30,34,54]			
SS-MVPA			[22,31]			[32]						
FMS-TPA			[5,46,47,50]		[49]	[21,24,44,46,53]	[49]		[22,25,26]			
LMS-TPA		[27]	[5,22,25,47]			[27,31,44,46]	[27]		[26,51]			
OCS-TPA	[27]		[5,27,46,47,51]			[31,44,46]	[27]		[22,25,26,34,51]			
SS-TPA		[27]	[22,27,31]				[27]		[26]			

Notes: FMS: fundamental movement skill; LMS: locomotor skill; OCS: object control skill; SS: stability skill; LPA: light physical activity; MVPA: moderate-vigorous physical activity; TPA: total physical activity; Not Sig.: not significant associations; Low Level: r/β = 0.10–0.29 or R^2^ = 1–8%; Moderate Level: r/β = 0.30–0.49 or R^2^ = 9–24%; Strong Level: r/β ≥ 0.50 or R^2^ ≥ 25%; −: negative associations.

**Table 5 ijerph-17-03566-t005:** Overall level of evidence for associations between FMS types and specific-intensity PA.

Bivariate	Not Sig.	Sig. (*n*)	Summary Coding	
			*n*/N (%)	Association (+, −, 0, ?)
FMS-LPA	[21,24,28,48]	[33] −	1/5 (20%)	+
LMS-LPA	[21,28,31,32]	[33] −	1/5 (20%)	+
OCS-LPA	[28,31,32]	[33] −	1/4 (25%)	?
SS-LPA	[31,32]	/	0	?
FMS-MVPA	[5,43,47,48,52]	[21,22,24,25,28,33,34,44,45,53,54]	11/16 (69%)	+
LMS-MVPA	[5,17,25,30,43,47,52]	[22,28,31,32,33,34,44,45,54]	9/16 (56%)	?
OCS-MVPA	[5,34,43,47,52]	[17,22,25,28,30,31,32,33,34,44,45,54]	12/17 (71%)	+
SS-MVPA	[22,31]	[32]	1/3 (33%)	?
FMS-TPA	[5,47,50]	[21,22,24,25,26,44,46,49,53]	9/12 (75%)	+
LMS-TPA	[5,22,25,47]	[26,27,31,44,46,51]	6/10 (60%)	+
OCS-TPA	[5,47]	[22,25,26,27,31,34,44,46,51]	9/11 (82%)	+
SS-TPA	[22,31]	[26,27]	2/4 (50%)	?

Notes: FMS: fundamental movement skill; LMS: locomotor skill; OCS: object control skill; SS: stability skill; LPA: light physical activity; MVPA: moderate-vigorous physical activity; TPA: total physical activity; Not Sig.: not significant associations; *n*: number of studies that reported a statistically significant association; N: number of studies that reported associations between the specific component of FMS and specific intensity of PA; +/−: positive/negative association, indicates ≥60% of studies and four or more studies supporting a significant association; 0: 0–33% of studies supporting a significant association; ?: indeterminate/inconsistent association, indicates 34–59% of studies and less than four studies supporting a significant association.

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
