# Peer review of "Relationship between Fundamental Movement Skills and Physical Activity in Preschool-aged Children: A Systematic Review"

_ijerph, 2020, doi:10.3390/ijerph17103566_

Round 1

Reviewer 1 Report

The analysis of the fundamental Movement Skills and Physical Activity in children of preschool age is very old, where in a brief survey we were able to find studies dating from 1774, where the interest was to find specific skills in children from an early age. The studies by Piaget, Ajuriaguerra, Gardner, among others demonstrate its importance over the years. The study is interesting but some factors are missing to clarify the reader. My assessment will be global in some points and specify in others.

The visualization of this theme in the last 10 years has shown a considerable amount of studies, where we are not able to understand why to go until 1995 to collect the 21 studies, and their own results demonstrate this, having only one reference from 1999, one from 2005 and one from 2008, being able to focus their study on the last 10 or even extend to 15 years (Since 2009 or 2004).

In the abstract and during the text, abbreviations appear that were not previously defined by readers (E.g. MPVA, TPA, OCS). It is important to define what they mean before using them, readers with less knowledge of this topic may not understand what it means.

In the introduction, care should be taken with the formatting of the text, (E.g. day[8, 9]) and there should be space (day [8, 9]). In some references in the text the end point after the reference should not appear, but flowing text (E.g. Crane et al [28]. and Iivonen et al [21].). Review the entire text.

Attention to English, sex is not written, but gender. Sometimes in the text there is sex and other gender, there must be a harmonization with the correct word.

The methodology is well described, but I call attention to the bibliographic reference referring to the statistical data because it does not show where it was published.

Something that brought some concern was that the data presented did not match correctly with a brief survey, where I found almost 9000 articles using their keywords following their inclusion and exclusion criteria.

Once again I call attention to the referencing in the text (E.g. Cohen’s (1988) [38]), the year of publication is not placed in this type of text referencing.

In the results, in the Methodological Quality of Studies, the authors indicate that the methodological scores are defined in the supplementary table S1 but in S1 there is only the indication of the quantity and place of search for the articles, wouldn't it be in S2?

After a good discussion, a good descriptive conclusion is expected, and not just a paragraph..

This theme has already been addressed several times in the journal, where not to present a single reference does not represent the real interest in the works published by the journal. Another factor is the fact that some references in your bibliography do not show the place where it was published. You need to review your references

I realize how hard it is to carry out such a study, but I recommend a review before it is approved.

Reviewer 2 Report

The autors have made a good work regarding fundamental movement skills and physical activity in preeschool-aged children but I have some concerns regarding the method:

For systematic reviews, the Cochrane Colaboration recommends to search in the Cochrane Database, WOS, and Ebsco. These motor search includes other databases as PubMed, SPORT discus, ERIC, etc. Autors are encouraged to carry out the search in the motor search recommended in order to find a higher quality results.

Authors used a tool to assess the methodological quality of their study but it is not clear wether it si validated or not, even though it was used by Holfelder and Schott. There are some validated scales to measure the methodological quality of sistematic reviews. Authors are encouraged to use a validated tool or the risk os bias assessment recommended by the Cochrane Colaboration.

In this kind of journals the systematic reviews should be completed with a meta-analisis. Authors justify that meta-analisis was not made because, following Ionnidis et al (2008), there were a considerable heterogeneity across the studies included. But Ionnidis also point out that forest plots should be presented even though the summary estimate is not calculated (the diamond at the bottom). Heterogeneity is usually calculated with I2. To excuse this incomplete analysis, authors should at least have included I2 scores. I recomment the authors to include the meta-analisis of this systematic review in order that this paper could be accepted for publication.

Reviewer 3 Report

The authors reviewed systematically published studies on the associations between basic/fundamental movement skills and physical activity indices in preschoolers aged 3-6 years. The authors concluded that the association between fundamental movement skills and physical activity indices was moderate, at maximum.

The aim of the systematic review is interesting, important and clearly stated.

Abstract: please spell-out abbreviations, when introduced for the first time such as TPA, OCS and LMS; without such fundamental information, the reader is lost.

“… while the 30 associations differ by gender, FMS subdomain….”; in my opinion, the sentence is not completed.

Introduction: “Sufficient physical activity (PA) is linked with healthy body weight [1, 2], cardiorespiratory and muscular fitness [2, 3], various cognitive effects [4], and fundamental movement skills (FMS) [5, 6].” While these statements appear to be true, please specify, among which samples such patterns of results could be observed. Otherwise, the reader might ask, what the present systematic review might be good for. For the assessment of basic motor skills, the authors might give a closer look at (Herrmann, Gerlach, & Seelig, 2015; Herrmann & Seelig, 2016; C. Herrmann, Heim, & Seelig, 2019)

Overall, the Introduction section was well written.

Methods: the authors described very well how they approached their tasks. The procedures are clearly reported such to allow an identical replication. Perhaps I did not read well, though, was language also introduced as inclusion-exclusion criterion?

Results: Perhaps, there was an issue in transforming the word-file into a pdf-file, though, is it possible that the Tables do not have any Notes to explain the abbreviations? Such Notes are particularly necessary for instance for Table 3, which remains completely enigmatic without further explanations.

Discussion: data and findings are very well discussed.

Conclusions: As often, it is a matter of taste, though, I suggest to shorten the Conclusion section; in contrast, do not use abbreviations, as often, readers focus on the Abstract and Conclusion section to grasp the most important information of a manuscript.

References

Herrmann, Gerlach, E., & Seelig, H. (2015). Development and validation of a test instrument for the assessment of basic motor competencies in primary school. . Measurement of  Physical Education Exercise Science, 19, 80-90.

Herrmann, & Seelig, H. (2016). MOBAK-5; Basic motor competencies in fifth grade: Testmanual.

Herrmann, C., Heim, C., & Seelig, H. (2019). Construct and correlates of basic motor competencies in primary school-aged children. Journal of sport and health science, 8(1), 63-70. doi:10.1016/j.jshs.2017.04.002

Round 2

Reviewer 1 Report

Dear authors

I appreciate the responses, as well as the changes made.

After analyzing the corrections and the document as a whole, I had a question due to the elimination of search engines like SPORTDiscus, Education Resources Information Center (ERIC), and Academic Search Premier?
Another question was related to the result of the quality score, as the author Barnett 2013 had a result of 7 in the first document and a result of 8 in the second document.
I understand that a study of this magnitude requires a great deal of time and, I reinforce the proposal to include more references from IJERPH, which has several studies on the subject that can be included in its theoretical framework or in its discussion. A single reference does not represent the real interest in the works published by the journal.

I realize how hard it is to carry out such a study, but I recommend a review before it is approved.

Best regards

Reviewer 2 Report

Thank you for revising and improving your work, however I still have some unclear points:

I dont agree with the author’s response 2. If there are either cross-sectional and longitudinal studies, their risk of bias should be analyzed with their respective scales: for example, the NOS (Wells GA, Shea B, O’Connell D, Peterson J, Welch V, Losos M, et al. The Newcastle-Ottawa Scale (NOS) for assessing the quality if nonrandomized studies in metaanalyses. Available from: URL: http://www.ohri.ca/programs/clinical_epidemiology/oxford.htm),) for the first ones, and the Cochrane recommendations for the second ones. If there are validated scales for assessing risk of bias, in my opinion, they should be used. On the other hand, the STROBE is not a tool for quality assessment. It only recommends the ítems that should be included when writting an observational study.

Regarding point 3, if authors still considar that meta-analisis can’t be implemented, the information regarding meta-analisis should be removed from the paper.
